# Downregulation of S6 Kinase and Hedgehog–Gli1 by Inhibition of Fatty Acid Synthase in AML with FLT3-ITD Mutation

**DOI:** 10.3390/ijms26125721

**Published:** 2025-06-14

**Authors:** Maxim Kebenko, Ruimeng Zhuang, Konstantin Hoffer, Anna Worthmann, Stefan Horn, Malte Kriegs, Jan Vorwerk, Nikolas von Bubnoff, Cyrus Khandanpour, Niklas Gebauer, Sivahari Prasad Gorantla, Walter Fiedler, Carsten Bokemeyer, Manfred Jücker

**Affiliations:** 1Hubertus Wald Tumorzentrum, Department of Oncology–Hematology, Bone Marrow Transplantation and Pneumology, University Cancer Center, 20246 Hamburg, Germany; fiedler@uke.de (W.F.); c.bokemeyer@uke.de (C.B.); 2Clinic for Hematology and Oncology, University Hospital Schleswig-Holstein Campus, 23562 Lübeck, Germanynikolaschristiancornelius.vonbubnoff@uksh.de (N.v.B.); cyrus.khandanpour@uksh.de (C.K.); niklas.gebauer@uksh.de (N.G.); sivahari.prasadgorantla@uksh.de (S.P.G.); 3Institute of Biochemistry and Signal Transduction, Center for Experimental Medicine, University Medical Center Hamburg-Eppendorf, 20246 Hamburg, Germany; ruimengzh@163.com (R.Z.); juecker@uke.de (M.J.); 4Department of Radiobiology & Radiation Oncology, UCCH Kinomics Core Facility, University Cancer Center Hamburg, University Medical Center Hamburg-Eppendorf, 20246 Hamburg, Germany; k.hoffer@uke.de (K.H.); m.kriegs@uke.de (M.K.); 5Department of Biochemistry and Molecular Cell Biology, University Medical Center Hamburg-Eppendorf, 20246 Hamburg, Germany; a.worthmann@uke.de; 6Research Department Cell and Gene Therapy, Department of Stem Cell Transplantation, University Medical Center Hamburg-Eppendorf, 20246 Hamburg, Germany; shorn@uke.de

**Keywords:** fatty acid synthase (FASN), palmitoylation, TVB-3166, AML, FLT3-ITD mutation, Akt, S6 kinase, MAPK, Gli1, Hedgehog signaling

## Abstract

Acute myeloid leukemia (AML) is a heterogeneous hematological malignancy associated with a poor prognosis. Activating mutations in the FLT3 gene occur in approximately 30% of AML cases, with internal tandem duplications in the juxtamembrane domain (FLT3-ITD; 75%) and mutations in the tyrosine kinase domain (FLT3-TKD; 25%). FLT3-ITD mutations are linked to poor prognosis and offer significant clinical predictive value, whereas the implications of FLT3-TKD mutations are less understood. The Hedgehog–Gli pathway is an established therapeutic target in AML, and emerging evidence suggests crosstalk between FLT3-ITD signaling and Gli expression regulation via non-canonical mechanisms. Post-translational modifications involving myristic and palmitic acids regulate various cellular processes, but their role in AML remains poorly defined. In this study, we investigated the role of fatty acid synthase (FASN), which synthesizes myristic and palmitic acids and catalyzes palmitoyl-acyltransferation, in regulating FLT3-ITD-Gli signaling. FASN knockdown using shRNA and the FASN inhibitor TVB-3166 was performed in FLT3-ITD-mutated AML cell lines (MOLM13, MV411) and Baf3-FLT3-ITD cells. The impact of FASN inhibition was assessed through Western blot and kinome profiling, while biological implications were evaluated by measuring cell viability and proliferation. FASN inhibition resulted in reduced levels of phospho-Akt (pAkt) and phospho-S6 kinase (pS6) and decreased expression of Hedgehog–Gli1, confirming non-canonical regulation of Gli by FLT3-ITD signaling. Combining TVB-3166 with the Gli inhibitor GANT61 significantly reduced the survival of MOLM13 and MV411 cells.

## 1. Introduction

Acute myeloid leukemia (AML) is a rare and aggressive hematological malignancy driven by various oncogenic mutations, including those affecting growth factor receptors, transcription factors, and chromatin–spliceosome complex modulators. Numerous recurrent chromosomal and molecular aberrations have significant prognostic and predictive implications in clinical practice. FLT3 mutations are the most common, occurring in approximately 30% of newly diagnosed adult AML patients [1,2]. Two major types of FLT3 mutations have been identified: internal tandem duplication (FLT3-ITD) within the juxtamembrane domain (75%) and mutations in the tyrosine kinase domain (FLT3-TKD) (25%). Additionally, FLT3-ITD mutations outside the juxtamembrane domain, such as the FLT3_ITD627E mutation, have been characterized. However, their biological and clinical characteristics are not yet fully understood [3]. Both mutations are constitutively active, leading to the downstream activation of pathways like STAT5, PI3K/Akt, and RAS/ERK [4,5]. FLT3-TKD activates signaling from the plasma membrane (PM), while FLT3-ITD primarily localizes to the endoplasmic reticulum (ER), activating the STAT5 pathway [2,6,7]. Furthermore, these mutations have distinct prognostic implications, with FLT3-TKD mutations typically associated with a better prognosis compared to FLT3-ITD mutations [8,9].

S-palmitoylation is a reversible post-translational modification (PTM) that plays a crucial role in the localization and function of many proteins, including receptor tyrosine kinases (RTKs). This modification is mediated by diverse palmitoyl acyltransferases (PATs), such as ZDHHC-type enzymes (ZDHHCs), as well as by fatty acid synthase (FASN), which is additionally responsible for the biosynthesis of palmitic and myristic acids. FASN activity is regulated by a variety of metabolic and growth factors through signaling pathways such as PI3K-Akt-mTOR and ERK [10,11,12]. Recent studies have highlighted the cysteine residue C563 as being essential for the palmitoylation of FLT3-ITD. Mutation of this residue (FLT3-ITD/C563S) disrupts palmitoylation, resulting in the re-localization of FLT3-ITD from the ER through the Golgi to the PM in a manner that is independent of palmitate. This shift alters FLT3-ITD signaling, favoring the PI3K/Akt and RAS/ERK pathways while maintaining STAT5 activation, likely through an indirect mechanism. Interestingly, this alteration in signaling promotes AML progression in mice, suggesting that targeting FLT3-ITD depalmitoylation could offer a promising therapeutic strategy, potentially in combination with FLT3 inhibitors [4,13].

In our study, we investigated the role of FASN in regulating FLT3-ITD and Hedgehog–Gli signaling in human and murine AML cell lines harboring FLT3-ITD mutations, based on the well-established crosstalk between these pathways, with the aim of exploring potential treatment combinations. Through kinome analysis and immunoblotting, we found that inhibiting FASN, via either shRNA or the FASN inhibitor TVB-3166, led to a significant downregulation of phospho-S6 (pS6), suggesting decreased p70S6 kinase activity. This effect was observed in human AML cell lines (MOLM13, MV411) and Baf3-FLT3-ITD-mutant cells. The reduction in pS6 was accompanied by a decrease in Gli1 expression, consistent with previous reports of non-canonical Gli1 regulation through mTOR/S6 signaling, as observed in esophageal adenocarcinoma cells. Biologically, the downregulation of pS6 and Gli1 correlated with a significant reduction in cell viability following TVB-3166 treatment. These findings suggest that FASN inhibition disrupts key signaling pathways critical for AML cell survival, underscoring the potential for therapeutic strategies combining FASN inhibitors with agents such as PAT, FLT3 inhibitors, or Hedgehog pathway inhibitors.

## 2. Results

### 2.1. Establishment of Stable FASN Knockdown in MOLM13 and MV411

Stable FASN knockdown was achieved in the FLT3-ITD-positive AML cell lines MOLM13 and MV411 using two distinct lentiviral vectors, kd1 and kd2. In MOLM13 cells, kd2 reduced FASN expression to 58% relative to the scrambled control (scr), while in MV411 cells, kd2 decreased FASN expression to 66% compared to scr (Figure 1A,B). Both reductions were statistically significant.

FASN knockdown by kd1 and kd2 was further confirmed in both cell lines through a biologically independent assay (Appendix A). To evaluate the biological effects of FASN knockdown, we conducted a fatty acid analysis using gas chromatography–mass spectrometry (GC-MS) with technical duplicates. In MOLM13 cells, FASN kd2 resulted in a reduction of palmitic acid levels from 100% to 90%, while in MV411 cells, kd1 and kd2 decreased myristic acid levels from 100% to 92% and 86%, respectively (Appendix A).

### 2.2. Reprogramming of FLT3-ITD Mutant Pathways and Downregulation of Gli1 in MOLM13, MV411, and Baf3-FLT3-ITDmut Cells upon FASN Inhibition

We first investigated the regulation of serine/threonine kinases by FASN through functional kinome analysis. To this end, FASN expression was reduced either by shRNA-mediated knockdown or by pharmacological inhibition using the FASN inhibitor TVB-3166. A significant downregulation of S6 and Akt was observed in MOLM-13 and MV4-11 cells following FASN knockdown with kd2, compared to the scrambled control (scr), as well as after treatment with 100 nM TVB-3166 (Figure 2 and Figure 3).

To validate these findings, immunoblotting assays were conducted. Although phospho-Akt (pAkt) levels could not be reliably quantified, a significant reduction in pS6 levels was observed in MV411 cells following FASN knockdown with kd1 (Figure 4A,B). In MOLM13 cells, pS6 levels were strongly reduced following kd2 knockdown; however, this decrease did not reach statistical significance (Appendix A). Notably, treatment with 100 nM TVB-3166 elicited a comparable reduction in pS6 levels, occurring 24 h post-treatment in MOLM13 cells and 48 h post-treatment in MV411 cells (Appendix A).

To further validate these findings, pS6 levels were evaluated at 0, 24, and 48 h in both FLT3-ITD-transduced and wild-type Baf3 cells following treatment with 100 nM TVB-3166 (Figure 5C). Successful transduction was confirmed by a significant increase in FLT3-ITD expression in transduced Baf3 cells compared to wild-type controls (Figure 5A). The treatment of Baf3-FLT3-ITD cells with TVB-3166 resulted in a significant reduction in pS6 levels at 24 h post-treatment. In contrast, no significant difference in pS6 levels was observed between untreated Baf3-FLT3-ITD cells and wild-type Baf3 cells (Figure 5B). Additionally, fluctuations in FLT3 and pS6 expression were observed in untreated cells over the 0–48 h time course, likely reflecting metabolic changes during the two-day incubation period (Figure 5C).

Furthermore, we investigated a potential link between FASN activity and the Hedgehog signaling pathway by analyzing the expression of Gli1. FASN knockdown in MOLM13 and MV411 cells led to a significant downregulation of Gli1, particularly with kd2 (Figure 6A,B). This downregulation was further confirmed in MOLM13 cells using a biologically independent assay following FASN knockdown with kd2 (Appendix A).

As previously reported, Kaosheng Lv et al. (2021) observed an upregulation of pFLT3 in MV411 cells upon disruption of FLT3-ITD palmitoylation through depletion of the palmitoyl acyltransferase ZDHHC6, which contributed to the progression of FLT3-ITD-mutant AML in mice [4]. To replicate this effect in our system, we evaluated pFLT3 levels in MOLM13 and MV411 cells following FASN inhibition, either by shRNA-mediated knockdown or by treatment with TVB-3166. Despite achieving significant FASN knockdown in both cell lines, no significant upregulation of pFLT3 was observed (Figure 7 and Figure 8).

Additionally, treatment with 100 nM TVB-3166 resulted in a marginal but statistically significant downregulation of pFLT3 in MOLM13 cells over a 0–48 h time course, compared to DMSO-treated controls (Figure 9A). In contrast, no significant changes in pFLT3 levels were observed in MV411 cells following TVB-3166 treatment (Figure 9B).

### 2.3. Reduction in the Viability of MOLM13 and MV411 Cells upon Combined Treatment with the FASN Inhibitor TVB-3166 and Gli 1/2 Inhibitor GANT61

The reported IC_50_ values for TVB-3166 range from 20 nM to 200 nM [12], whereas the IC_50_ for GANT61, a Gli1 inhibitor that blocks DNA binding, is approximately 5 μM [14]. In our study, TVB-3166 significantly reduced the cell viability of both MOLM13 and MV411 cells at concentrations between 100 nM and 250 nM. In contrast, lower concentrations (10 nM and 25 nM) had no significant effect on the cell viability of either cell line compared to treatment with dimethylsulfoxide (DMSO). To validate the impact of FASN inhibition, cell viability was also assessed in stable FASN-knockdown cell lines (kd1 and kd2), with no significant changes observed in either MOLM13 or MV411 cells (Figure 10A,B and Appendix A). Similarly, GANT61 treatment at concentrations of 5 μM and 10 μM did not significantly affect the cell viability of MOLM13 or MV411 cells. However, co-treatment with 25 nM TVB-3166 and 10 μM GANT61 led to a marked and statistically significant reduction in cell viability for both cell lines (Figure 10A,B).

We also evaluated the cell proliferation of MOLM13 and MV411 cells following FASN inhibition via shRNA or treatment with TVB-3166. Neither genetic nor pharmacological inhibition of FASN resulted in significant changes in the proliferative activity of either cell line (Appendix A).

## 3. Discussion

FLT3 mutations are among the most frequent genetic alterations in adult acute myeloid leukemia (AML) and are associated with heterogeneous clinical outcomes. These mutations occur predominantly as internal tandem duplications (FLT3-ITD, ~75%) or point mutations within the tyrosine kinase domain (FLT3-TKD, ~25%), each activating distinct downstream signaling cascades [1,7]. FLT3-ITD is typically linked to the cytoplasmic tyrosine kinase Lyn and preferentially drives STAT5 activation, whereas FLT3-TKD predominantly activates the PI3K/Akt and Src/JAK/STAT3 pathways independently of Lyn involvement [15,16].

Recent work by Kaosheng Lv et al. has highlighted the role of S-palmitoylation in modulating FLT3-ITD signaling. Specifically, palmitoylation at cysteine residue C563, catalyzed by the palmitoyl acyltransferase ZDHHC6, was shown to be critical for FLT3-ITD-mediated STAT5 activation in AML cells harboring FLT3-ITD mutations. Intriguingly, inhibition of FLT3-ITD palmitoylation led to its retention in the ER, followed by trafficking through the Golgi apparatus to the PM. This palmitate-independent localization reprogrammed FLT3-ITD signaling, resulting in constitutive activation of the Akt, ERK, and STAT5 pathways, thereby promoting leukemogenesis in vivo [4].

In this study, we investigated the role of fatty acid synthase (FASN) in modulating FLT3-ITD signaling. Unlike the ZDHHC family of palmitoyl acyltransferases (PATs), which specifically catalyze the palmitoylation of proteins, FASN is involved in both the de novo synthesis of palmitic acid and the broader regulation of protein palmitoylation through its metabolic products. To evaluate the impact of FASN on FLT3-ITD signaling, we generated stable FASN knockdowns via shRNA in two human leukemia cell lines, MOLM13 and MV411.

FASN depletion resulted in a significant reduction in intracellular levels of palmitic and myristic acids. Interestingly, despite this metabolic alteration, FLT3 phosphorylation remained unchanged. This finding aligns with those of a study by Kaosheng Lv et al. [4], which showed that FLT3-ITD retains its PM localization and kinase activity independent of palmitic acid following its re-localization from the ER upon inhibition of protein palmitoylation. However, a further analysis of downstream signaling revealed a marked reduction in pS6 in both MOLM13 and MV411 cells after FASN knockdown or pharmacological inhibition using the FASN inhibitor TVB-3166. A similar decrease in pS6 levels was observed in Baf3 cells expressing FLT3-ITD upon TVB-3166 treatment. This FASN-dependent downregulation of p70S6 kinase activity contrasts with the upregulation of pAkt reported by Lv et al. Notably, our findings are consistent with those of Matias Blaustein et al. and Shasha Yin et al. [17,18], who demonstrated that the palmitoylation status directly influences the phosphorylation and subcellular localization of upstream regulators such as Akt and mTOR. Given that p70S6 kinase is a direct downstream target of the Akt–mTOR signaling axis, we propose that reduced FASN activity compromises Akt activation, thereby attenuating mTOR phosphorylation and resulting in diminished S6 phosphorylation.

The divergence between our findings and those of Kaosheng Lv et al. may stem from differences in experimental design. Their study exclusively focused on inhibiting protein palmitoylation without simultaneously targeting the synthesis of palmitic acid. Moreover, they did not evaluate the expression or potential compensatory functions of other PATs, nor did they assess the palmitoylation status of key signaling mediators such as Akt, mTOR, ERK, and STAT5 following ZDHHC6 inhibition. It is therefore plausible that compensatory mechanisms, potentially involving other ZDHHC family members or metabolic enzymes like FASN, sustain the palmitoylation—and, thereby, the phosphorylation and activity—of these signaling proteins in the absence of ZDHHC6 activity.

Furthermore, we investigated whether FASN activity influences the expression of Gli1, a key downstream effector of the Hedgehog pathway, in our AML cell models. This inquiry is based on the well-established non-canonical activation of Gli downstream of FLT3-ITD signaling in AML, as well as in other malignancies [19,20,21,22]. Additionally, the correlation between elevated Gli expression and poor prognosis in AML, along with the clinical approval of Glasdegib—a Hedgehog–Smoothened (SMO) inhibitor—for AML treatment [23], underscores the rationale for exploring the combined targeting of FLT3, FASN, and Gli. We observed a consistent downregulation of Gli1 in both AML cell models, MOLM13 and MV411, following FASN inhibition via shRNA. This finding aligns with the established direct regulation of Gli expression by S6 kinase, as demonstrated, for example, in esophageal adenocarcinoma [14].

Finally, we evaluated the functional impact of our findings by assessing cell viability and proliferation following treatment with the FASN inhibitor TVB-3166 and the GLI inhibitor GANT61, as well as in stable FASN-knockdown MOLM13 and MV411 cells. When applied individually at lower concentrations (10–25 nM for TVB-3166; 5–10 µM for GANT61), neither inhibitor significantly affected cell viability. However, their combined application at these concentrations resulted in a marked and statistically significant reduction in cell viability, indicating a potential synergistic effect. Interestingly, stable knockdown of FASN alone did not lead to significant changes in cell viability, nor did we observe substantial differences in proliferation in either cell line following treatment with TVB-3166 or FASN shRNA. The absence of an effect in FASN-knockdown cells may be attributed to the establishment of a new cellular steady state under prolonged knockdown conditions, potentially influenced by extended puromycin selection.

The lack of proliferation impairment despite inhibition of the PI3K/Akt signaling pathway has been previously reported. For example, treatment of mammary tumor cell lines with the FGFR tyrosine kinase inhibitor TKI258 significantly decreased cell survival via apoptosis induction yet did not alter proliferation marker expression [24]. The authors suggested that the pro-apoptotic activity of TKI258 was insufficient to override concurrent growth-promoting signals. An alternative explanation may involve the paradoxical role of apoptotic signaling in tumor biology, where apoptotic stimuli can, under certain conditions, enhance tumor proliferation. This paradox has been observed in several tumor models, and elevated levels of apoptosis have been associated with poor clinical outcomes in various cancers [25,26,27].

## 4. Materials and Methods

### 4.1. Materials and Reagents

#### 4.1.1. Antibodies

Monoclonal antibodies against pFLT3 (#3464S), pan-Akt (#4685S), pAkt S473 (#4060S), S6 (#2217S), pS6 (#2215S), MAPK (#4695S), and pMAPK (#4377S) were purchased from Cell Signaling Technology (Beverly, MA, USA). Antibodies targeting FASN (#48357), FLT3 (#sc-479), and Gli-1 (#515751) were obtained from Santa Cruz Biotechnology (Heidelberg, Germany). HRP-conjugated secondary antibodies against mouse IgG (#7076) and rabbit IgG (#7074) were obtained from Cell Signaling Technology.

#### 4.1.2. Vectors

PLKO.1-puro vectors encoding either FASN-targeting or non-targeting scrambled (scr) shRNA were obtained from Sigma-Aldrich (Taufkirchen, Germany). The human FLT3-ITD gene was transferred from pMys-IG-FLT3-ITD (R1079), a kind gift from Carol Stocking (Leibniz Institute of Virology), into the third-generation lentiviral vector LeGO-iB2/Zeo via Not I cloning. Sequence identity was confirmed by Sanger sequencing.

#### 4.1.3. Inhibitors

TVB-3166 was developed and kindly provided by Sagimet Biosciences (formerly 3-V Biosciences, San Mateo, CA, USA).

### 4.2. Methods

#### 4.2.1. Culturing of Cells

The following cell lines, all registered in the ExPASy Cellosaurus database, were used: MOLM13 (RRID:CVCL_2119), MV411 (RRID:CVCL_0064), HEK293 (RRID:CVCL_0045), and Baf3 (RRID:CVCL_0161). Cells were obtained from the German Collection of Microorganisms and Cell Cultures GmbH (DSMZ) and authenticated using the Multiplexion test. All cell lines were confirmed to be mycoplasma-free through routine testing.

Cells were maintained at a density of 3 × 10^5^ to 3 × 10^6^ viable cells/mL and counted using trypan blue exclusion with a Neubauer chamber. MOLM13 and MV411, two human FLT3-ITD-mutant AML cell lines, were cultured in Roswell Park Memorial Institute (RPMI)-1640 medium supplemented with 20% fetal calf serum (FCS) and 1% penicillin/streptomycin (P/S). Parental Baf3 and control cells were cultured in RPMI-1640 medium containing 10% FCS, 1% P/S, 1% glutamine, and 5 ng/mL recombinant murine IL-3. Baf3-FLT3-ITD cells, transduced with LeGO-FLT3-ITD-iB2/Zeo and expressing mTagBFP and Zeocin resistance, were maintained without IL-3. HEK293T cells were cultured in DMEM with 10% FCS and without P/S. All cells were handled in a class II safety cabinet and incubated at 37 °C with 5% CO_2_.

#### 4.2.2. Transformation and Plasmid Preparation

A total of 100 μL XL1-Blue competent cells were transformed with 100 ng plasmid DNA, following the manufacturer’s protocol. Plasmid DNA was purified using the NucleoBond Xtra kit (Sigma-Aldrich, Taufkirchen, Germany). Additional protocol details are provided in the Appendix A.

#### 4.2.3. Lentiviral Knockdown of FASN

PLKO.1-puro vectors encoding FASN-shRNA or scrambled shRNA were purchased from Sigma-Aldrich. Two knockdown clones (kd1 and kd2) were generated to confirm their reproducibility. HEK293T cells were plated at a density of 2 × 10^5^ cells per 10 cm dish. For transfection, 2.5 μg vector DNA and 20 μL P3000 reagent were combined, followed by 8 μg each of VSVG, gagPol, and HIV1-Rev plasmids. Lipofectamine was used for transfection according to the manufacturer’s instructions.

Target cells were seeded at 3 × 10^5^ cells/well in RPMI medium. Viral supernatant was collected and added at 24 h and 48 h post-transfection. Selection was conducted using 4 μg/mL puromycin until control (untransduced) cells were no longer viable. All lentiviral work was performed under biosafety level 2 (BSL-2) conditions with institutional approval in accordance with German law.

#### 4.2.4. Immunoblotting

Protein lysates were prepared using NP40 buffer (BostonBioProducts, Milford, MA, USA), and concentrations were measured using the DC protein assay kit (Bio-Rad, Munich, Germany). Proteins were separated on 4–20% SDS-PAGE gels (Thermo Fisher Scientific, Waltham, MA, USA) and transferred to nitrocellulose membranes (Amersham, GE Healthcare) at 65 V for 2 h. Membranes were stained with Ponceau and cut as required, then incubated overnight at 8 °C with primary antibodies (1:1000 dilution). After washing, HRP-conjugated secondary antibodies (1:5000 dilution) were added for 1 h at room temperature. Blots were developed using the LAS-4000 imaging system and SuperSignal West Dura substrate (Thermo Fisher Scientific, Waltham, MA, USA).

#### 4.2.5. Fatty Acid Analysis

Fatty acids were analyzed by gas chromatography–mass spectrometry (GC–MS). Cell pellets were resuspended in 50 μL water with 100 μL internal standard mix (tetradecanoate-d27 and heptadecanoate-d33, 200 μg/mL in methanol/toluene 4:1), then 1000 μL methanol/toluene (4:1) and 100 μL acetyl chloride were added. Samples were vortexed and heated at 100 °C for 1 h. After cooling, 3 mL of 6% sodium carbonate was added, followed by centrifugation (1800× *g*, 5 min). The upper organic layer was transferred to autosampler vials.

Analyses were performed using a Trace 1310 GC (Thermo Fisher Scientific, Waltham, MA, USA) equipped with a DB-225 column (30 m × 0.25 mm i.d., 0.25 μm film; Agilent) and coupled to an ISQ 7000 mass spectrometer in selected ion monitoring (SIM) mode. Chromeleon 7 software was used for data acquisition. Peaks were identified and quantified by comparison to internal standards and known reference chromatograms.

#### 4.2.6. Functional Kinome Profiling

Kinome profiling was conducted as described previously [24] using the PamStation^®^12 platform and PTK-PamChip^®^ arrays (PamGene International, HH’s-Hertogenbosch, the Netherlands). Whole-cell lysates were prepared in M-PER buffer with protease and phosphatase inhibitors (1:100 dilution each; Pierce, CO, USA), at 100 μL per 1 × 10^6^ cells, and stored at –80 °C. Protein concentrations were measured using the BCA assay (Merck KGaA, Darmstadt, Germany). One microgram of protein and 400 μM ATP were applied per chip. Peptide phosphorylation was detected using a fluorescein-conjugated PY20 antibody (Exalpha, Boston, MA, USA), imaged using a CCD camera. Data were analyzed using Evolve and BioNavigator v6.3.67.0 software (PamGene, Thousand Oaks, CA, USA).

#### 4.2.7. Cell Viability Assay

Cell viability was assessed using the trypan blue exclusion assay and a Vi-CELL™ XR analyzer (Beckman Coulter, Krefeld, Germany). MOLM13 and MV411 cells (untransfected or expressing scr/FASN shRNA) were treated for 5 days with DMSO (control), TVB-3166 (10–250 nM), GANT61 (5–50 μM), or combinations thereof. Cell counts were recorded post-incubation.

#### 4.2.8. Proliferation Assay

For proliferation studies, 10,000 MOLM13 or MV411 cells were plated per well in 96-well plates with 100 μL RPMI containing 20% FCS and 1% P/S. After 24 h, 100 μL of inhibitor solution was added. Cell confluence was monitored using the IncuCyte Zoom system (Essen BioScience, Ann Arbor, MI, USA). FASN-knockdown and scrambled control cells were plated in 200 μL RPMI with 20% FCS, 1% P/S, and 1.5 μg/mL puromycin.

### 4.3. Statistical Analysis

The normality of the data was assessed using the Kolmogorov–Smirnov test with Lilliefors correction, the Shapiro–Wilk test, and the Anderson–Darling test. Western blot data were primarily analyzed using technical triplicates or quadruplicates, whereas cell viability and proliferation assays were conducted using biological triplicates. Statistical significance was evaluated using one-way ANOVA followed by Tukey’s post hoc test for experiments involving more than two groups, with a minimum sample size of n = 3. For comparisons involving two groups or when the sample size was n < 3, unpaired Student’s *t*-tests were applied. Paired *t*-tests were used to evaluate specific pairwise comparisons within experiments comprising multiple data groups. Significance levels are indicated in the figures as *p* < 0.05 (*). For datasets exhibiting high variance, Cohen’s d was calculated to assess the effect size, with a large effect defined as d = 0.8–1.2 and indicated by a hash symbol (#). Error bars represent standard deviations (SDs). To account for differences in protein expression, phosphorylated-to-total protein ratios were calculated to independently assess phosphorylation levels. For improved standardization in Western blot quantifications, most analyses were normalized to total protein using HSC70 or β-actin as the loading control.

Software

Microsoft Office 2007

Graphpad Prism: Version 8.2

AIDA Image Analyzer: Version 3.44

Zotero: 6.0.20

## 5. Conclusions

Inhibition of FASN—a central enzyme in both de novo palmitic acid synthesis and protein palmitoylation—using the selective inhibitor TVB-3166 resulted in downregulation of the Akt–S6 signaling axis and reduced expression of the transcription factor Gli1, thereby significantly impairing the survival of AML cells. These findings underscore the pivotal regulatory role of FASN in oncogenic signaling pathways relevant for leukemia cell survival.

To better understand the mechanistic underpinnings of FASN-mediated signaling, further studies are warranted to assess the palmitoylation status and subcellular localization of critical signaling molecules such as Akt, mTOR, and p70S6 kinase in AML cells following FASN inhibition. Additionally, the observed lack of a pronounced anti-proliferative effect of TVB-3166 in leukemia cells necessitates deeper investigations into potential resistance mechanisms. Elucidating these compensatory survival pathways could provide the basis for rational combinatorial treatment strategies that enhance the efficacy of FASN-targeted therapy.

Given that several FASN inhibitors are currently undergoing clinical evaluation, future research should also explore the therapeutic potential of combination regimens involving FASN inhibitors with Hedgehog or FLT3 inhibitors in FLT3-ITD-mutated AML [27,28].

## Figures and Tables

**Figure 1 ijms-26-05721-f001:**
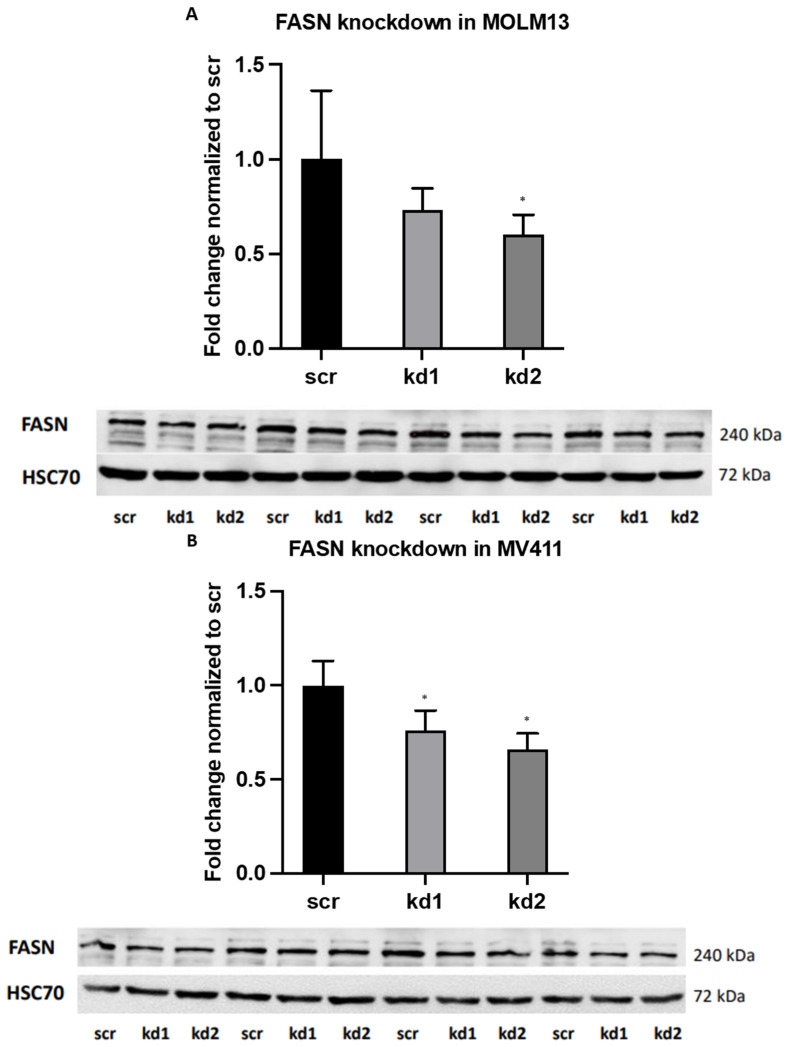
shRNA-mediated knockdown of endogenous FASN in MOLM13 (**A**) and MV411 (**B**). MOLM13 and MV411 cell lines were transduced with FASN-targeting shRNA constructs (vector 1, designated kd1, or vector 2, designated kd2) or a non-targeting control vector (scr) to assess baseline FASN expression. Protein lysates were collected and analyzed by Western blot in technical quadruplicates, using HSC70 as a loading control for FASN quantification. FASN expression levels in kd1 and kd2 cells were normalized to those in scr controls. A significant reduction in FASN expression was observed with kd2 in both MOLM13 (**A**) and MV411 (**B**) cells. Densitometric quantification of phospho-protein/protein ratios was conducted following normalization to scr controls, with results presented as means ± standard deviations. Statistical significance was determined using one-way ANOVA followed by Tukey’s post hoc test (* *p* < 0.05).

**Figure 2 ijms-26-05721-f002:**
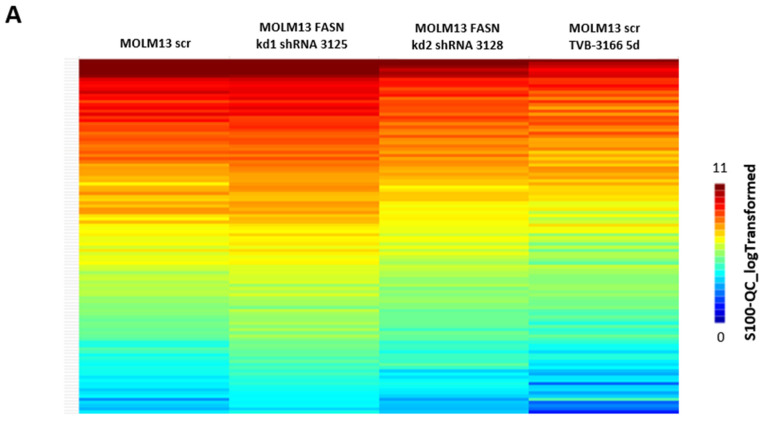
An analysis of FLT-ITD signal transduction in MOLM13 by PamGene (serine/threonine kinases). Serine/threonine kinases were analyzed in MOLM13 cells with FASN knockdowns kd1 and kd2 or after treatment with 100 nM TVB-3166 using functional kinome profiling. The heatmap displays the 110 analyzed peptides, with the S100_log-transformed values shown (**A**). A volcano plot highlights significantly altered peptides (scr control vs. treatment), where the x-axis represents the log fold change in peptide phosphorylation (dashed line = 0), and the y-axis shows the significance (–log *p*-value) for each peptide (**B**). Significant changes are indicated by a threshold of >1.3 (dashed/dotted line). The upstream kinase analysis compares scr vs. kd2 and scr vs. TVB-3166. A normalized kinase statistic (log2) of <0 indicates lower kinase activity in the inhibitor-treated sample, while a specificity score (log2) of >1.3 (represented by white to red circles) denotes statistically significant changes (**C**).

**Figure 3 ijms-26-05721-f003:**
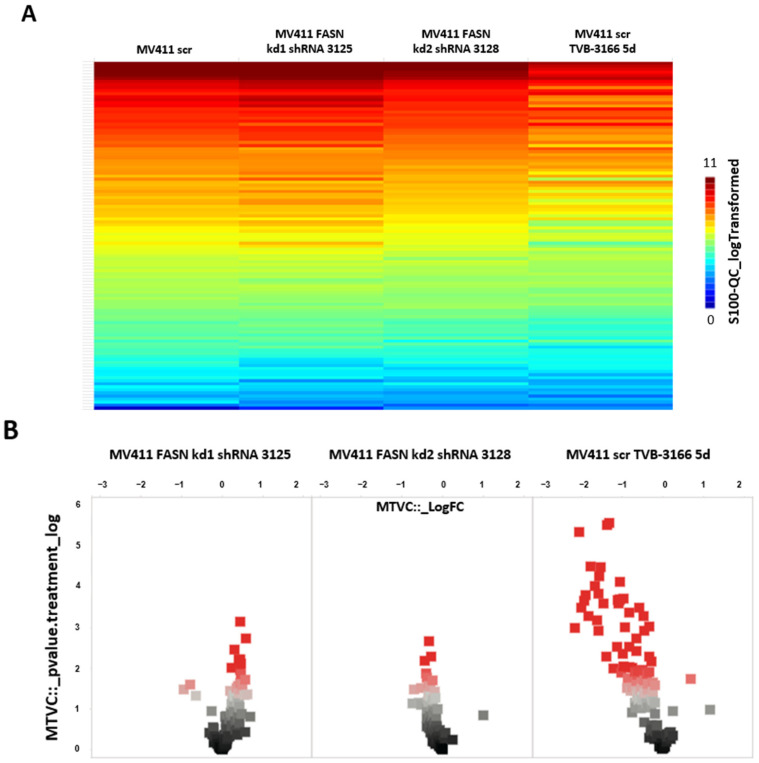
An analysis of FLT-ITD signal transduction in MV411 by PamGene (serine/threonine kinases). Serine/threonine kinases were analyzed in MV411 cells with FASN knockdowns (kd1 and kd2) or after treatment with 100 nM TVB-3166 using functional kinome profiling. The heatmap displays the 110 analyzed peptides, with the S100_log-transformed values shown (**A**). A volcano plot highlights significantly altered peptides (scr control vs. treatment), where the x-axis represents the log fold change in peptide phosphorylation (dashed line = 0), and the y-axis shows the significance (–log *p*-value) for each peptide. Significant changes are indicated by a threshold of >1.3 (dashed/dotted line) (**B**). The upstream kinase analysis compares scr vs. kd2 and scr vs. TVB-3166. A normalized kinase statistic (log2) of <0 indicates lower kinase activity in the inhibitor-treated sample, while a specificity score (log2) of >1.3 (represented by white to red circles) denotes statistically significant changes (**C**).

**Figure 4 ijms-26-05721-f004:**
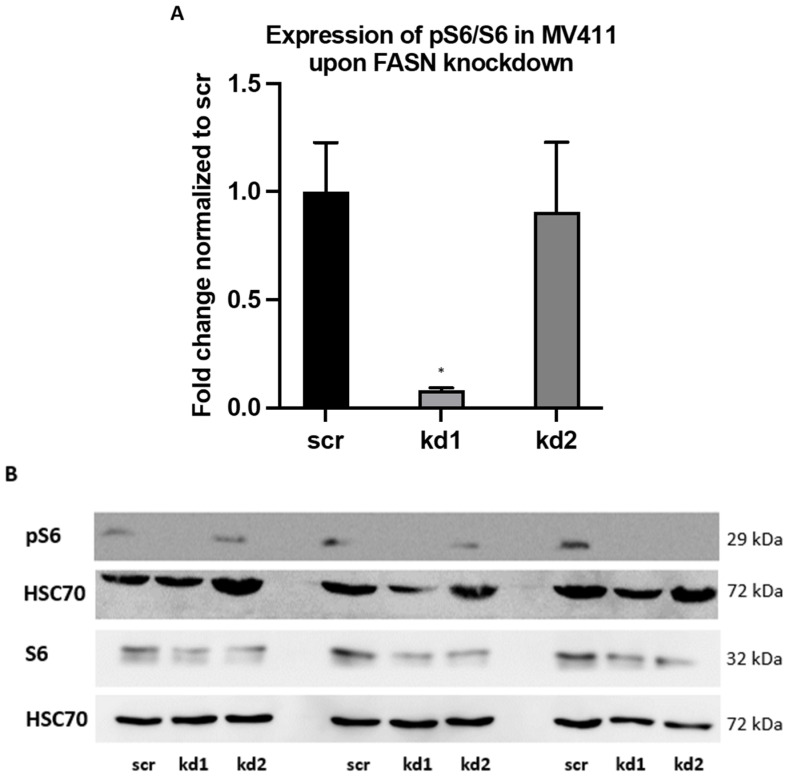
An analysis of pS6 expression in FASN-knockdown MV411 cells by Western blot (**A**,**B**). Protein lysates from MV411 cells were analyzed to evaluate the expression of pS6 and total S6 (S6) in technical triplicates for each cell line, with HSC70 used as a normalization control. The expression levels of pS6 and S6 in kd1 and kd2 were normalized to those in scr. In MV411 cells, stable knockdown of FASN kd1 (shown in Figure 1B) led to a reduction in pS6 expression compared to scr. Densitometric quantification of the phospho-protein/protein ratios was conducted after normalization to scr, with the results presented as mean values and standard deviations. Statistical significance was determined using one-way ANOVA followed by Tukey’s post hoc test (* *p* < 0.05).

**Figure 5 ijms-26-05721-f005:**
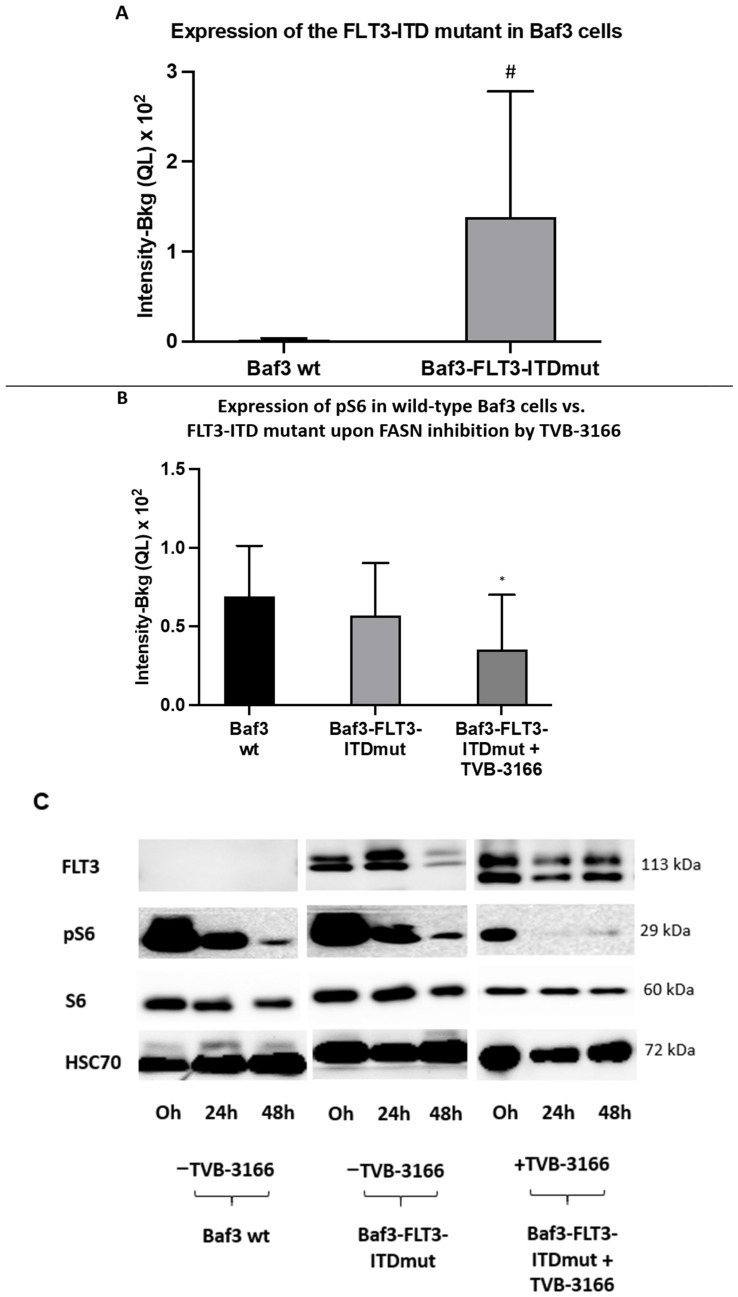
An analysis by Western blot of pS6 expression in FLT-ITD-mutated Baf3 cells treated with TVB-3166. (**A**) Significantly higher FLT3-ITD expression levels were observed in transduced Baf3-FLT3-ITDmut cells compared to wild-type (wt) Baf3 cells. Due to notable variability between the two groups, the magnitude of the difference was quantified using Cohen’s *d*, yielding a large effect size (Cohen’s *d* = −1.10; indicated by #). (**B**) pS6 and S6 expression levels remained unchanged in untreated Baf3-FLT3-ITDmut cells compared to Baf3 wild-type controls. However, treatment with 100 nM TVB-3166 resulted in a significant reduction in pS6 expression in Baf3-FLT3-ITDmut cells, while S6 expression remained unaffected. Given the small sample size, statistical significance was assessed separately for the pS6/HSC70 and S6/HSC70 ratios using unpaired Student’s *t*-tests (*p* < 0.05; indicated by *). (**C**) Protein lysates were collected at 0, 24, and 48 h (n = 3) to assess FLT3-ITD expression and at 0 and 24 h (n = 2) to evaluate pS6 and total S6 expression in both wild-type Baf3 and Baf3-FLT3-ITDmut cells. A densitometric analysis of phospho-to-total protein ratios was performed following normalization to HSC70. Data are presented as means ± standard deviations across the indicated time points.

**Figure 6 ijms-26-05721-f006:**
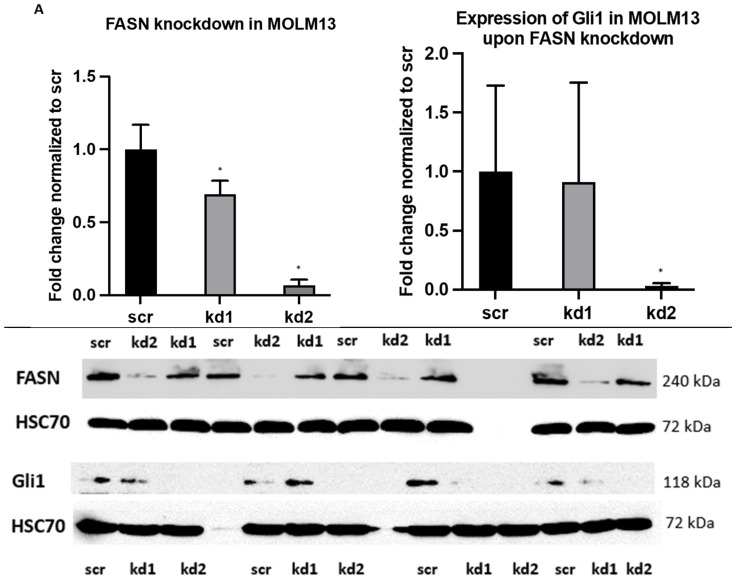
An analysis of Gli1 expression in MOLM13 and MV411 by Western blot. Protein lysates from MOLM13 and MV411 cells were analyzed to assess the expression of Gli1 in technical quadruplicates for each cell line, with HSC70 serving as a normalization control. The expression levels of Gli1 in kd1 and kd2 were normalized to those in the scr control. (**A**) In kd2 MOLM13 cells, stable knockdown of FASN resulted in decreased Gli1 expression compared to scr. (**B**) Similarly, in kd2 MV411 cells, stable knockdown of FASN (shown in Figure 1B) led to reduced Gli1 expression compared to scr. Densitometric quantification of the phospho-protein/protein ratios was performed after normalization to scr, and the results are presented as mean values with standard deviations. Statistical significance was determined using one-way ANOVA followed by Tukey’s post hoc test (* *p* < 0.05).

**Figure 7 ijms-26-05721-f007:**
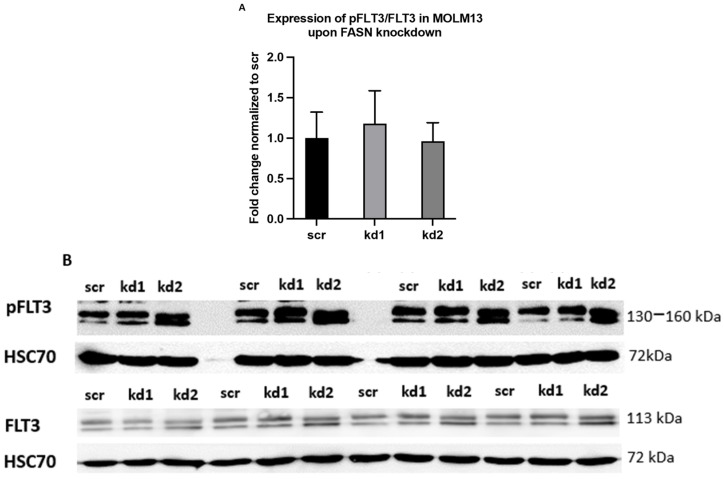
An analysis of pFLT3 expression in FASN-knockdown MOLM13 cells by Western blot. (**A**) Highly significant FASN knockdown was detected in kd2 MOLM13 cells (shown in Figure 6A). However, no significant changes in pFLT3 levels were observed in FASN-knockdown cells compared to scr. Statistical significance was determined using one-way ANOVA followed by Tukey’s post hoc test. (**B**) Protein lysates were analyzed to evaluate the expression of FASN, pFLT3, and total FLT3 (FLT3) in technical quadruplicates in MOLM13 cells, with HSC70 used as a normalization control. The expression levels of the target proteins in kd1 and kd2 were normalized to those in the scr control. Densitometric quantification of the phospho-protein/protein ratios was performed after normalization to scr, and the results are presented as mean values with standard deviations.

**Figure 8 ijms-26-05721-f008:**
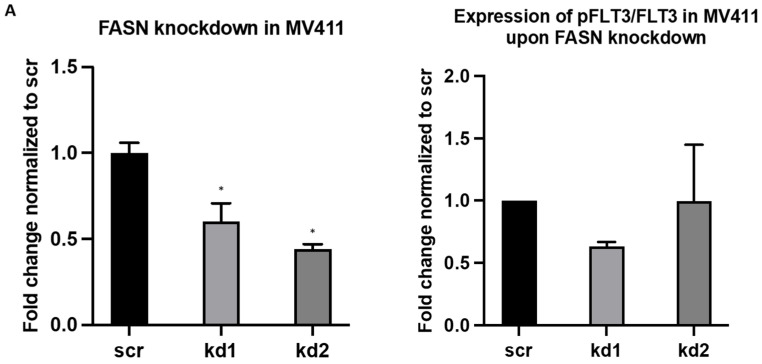
An analysis of pFLT3 expression in FASN-knockdown MV411 cells by Western blot. (**A**) Significant FASN knockdown was detected in kd1 MV411 cells. No significant changes in pFLT3 levels were observed in FASN-knockdown MV411 cells compared to scr. Densitometric quantification of the phospho-protein/protein ratios was performed after normalization to scr, and the results are presented as mean values with standard deviations. Statistical significance was determined using one-way ANOVA followed by Tukey’s post hoc test (* *p* < 0.05). (**B**) Protein lysates were analyzed to evaluate the expression of FASN, pFLT3, and total FLT3 (FLT3) in technical quadruplicates for MV411 cells, with HSC70 used as a normalization control. The expression levels of the target proteins in kd1 and kd2 were normalized to those in the scr control.

**Figure 9 ijms-26-05721-f009:**
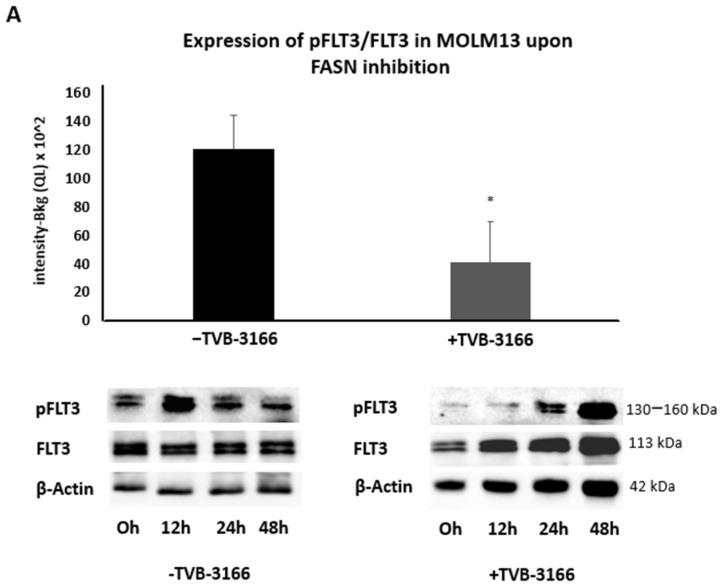
An analysis of pFLT3 expression in TVB-3166-treated MOLM13 and MV411 cells by Western blot. Protein lysates were analyzed to assess the expression of pFLT3 and total FLT3 (FLT3) in MOLM13 and MV411 cells, with β-actin used as a normalization control. Data were collected at 0, 4, 12, 24, and 48 h and are presented as mean values across these time points. For improved visualization, the Western blots for untreated MOLM13 (**A**) and MV411 (**B**) cells were combined by omitting the 4 h time point in each blot. (**A**) Treatment with 100 nM TVB-3166 led to a marginal downregulation of pFLT3 expression in MOLM13 cells. (**B**) No significant changes in pFLT3 levels were observed in MV411 cells following treatment with 100 nM TVB-3166. Densitometric quantification of phospho-protein-to-total protein ratios was performed following normalization to β-actin. The results are presented as means ± standard deviations. Statistical significance was determined using unpaired Student’s *t*-tests, with *p* ≤ 0.05 considered significant and indicated by *.

**Figure 10 ijms-26-05721-f010:**
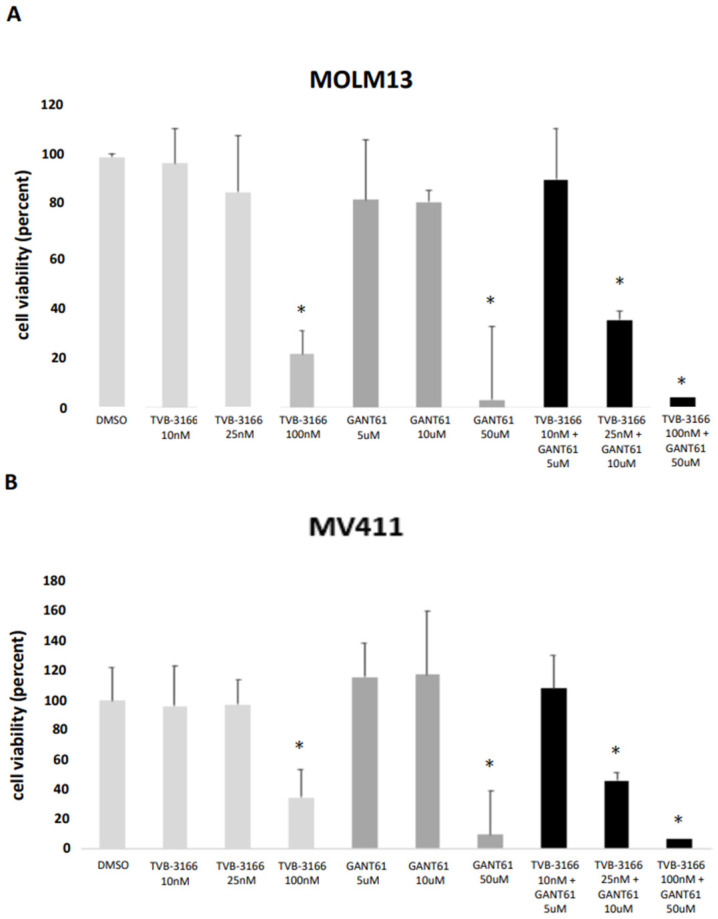
The cell viability of MOLM13 and MV411 cells upon treatment with FASN inhibitor TVB-3166 and Gli1 inhibitor GANT61. Cell viability was assessed on day 5 following treatment with the inhibitors using the Vi-CELL™ XR system and a trypan blue exclusion assay. The experiments were performed in biological triplicates for each cell line. Statistical significance was initially assessed using one-way ANOVA, followed by Tukey’s post hoc test for multiple group comparisons. In addition, paired *t*-tests were used to evaluate specific pairwise comparisons, including “DMSO vs. TVB-3166 (25 nM)”, “DMSO vs. GANT61 (10 μM)”, and “DMSO vs. TVB-3166 (25 nM) + GANT61 (10 μM)”, among others (* *p* < 0.05). Single-agent treatments with TVB-3166 (10 nM, 25 nM) or GANT61 (5 μM, 10 μM) did not significantly affect the cell viability of either MOLM13 (**A**) or MV411 (**B**) cells. However, co-treatment with TVB-3166 (25 nM) and GANT61 (10 μM) led to a significant reduction in cell viability compared to TVB-3166 monotherapy, indicating a potential additive effect.

## Data Availability

Primary data of the experiments in this manuscript are by the senior, and by the first author.

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
