# Peer review of "Downregulation of S6 Kinase and Hedgehog–Gli1 by Inhibition of Fatty Acid Synthase in AML with FLT3-ITD Mutation"

_ijms, 2025, doi:10.3390/ijms26125721_

Round 1

Reviewer 1 Report

Comments and Suggestions for Authors

The manuscript "Downregulation of S6 kinase and Hedgehog-Gli1 by inhibition of fatty acid synthase in AML with FLT3-ITD mutation" shows interesting information but the observed results are not always consistent with the densitometry and the conclusions claimed by the authors.

  • Figure legends must be added for all figures. In the case of Figure 2, add them since Figures 2 and 3 appear together.
  • The Western blots in Figure 4A are not clear; in fact, there are some bubbles in the bands. I recommend looking for better ones.
  • Figure 5B, be consistent about using numbers or asterisks to display p values
  • In Figure 5, why is pS6 phosphorylation also reduced over time in Baf3 wt?
  • The western blots in Figure 6 do not correspond with the densitometry; the reduction in Gli1 is not seen as mentioned.
  • Figure 6 be consistent how samples kd1 and k2 are placed in all western blots
  • Figures 6 and 9 are not consistent with western blots and densitometries.
  • What statistical tests did you use to determine that your data were parametric? Which ones did you use to make comparisons between groups? Your statistical description should be extensively supplemented.

Author Response

Dear Reviewer 1,

We would like to sincerely thank you for your review of our manuscript. Please find below our point-by-point response to your comments. We hope that the revisions we have made address your concerns satisfactorily in this revised version.

Best regards,

Maxim Kebenko

Reviewer 2 Report

Comments and Suggestions for Authors
  1. The organization of this work should be improved carefully. The results and discussion sections should be improved.
  2. What are the symbols *, **, and *** represented?
  3. Please improve the presentation of the figures.
  4. The images from Page 4 and Page 5 should be marked as Figure 2.
  5. It should give the titles and annotations to the figures.
  6. For the fatty acid analysis by gas chromatography, the GC-MS spectrum could be provided as SI.
  7. It should mark the molecular weight of the bands from the blotting results.
  8. Do not start a sentence with numbers. See the text from section “4.2.2. Transformation and plasmid preparation”.
  9. What’s RPMI medium?
  10. Please check the writing and expression carefully. There should be one space between numbers and letters (See, 100μl and 100ng). Please check the text “20 μl”, “2 ml”, and “4 μg/ml”, and it should be “20 μL”, “2 mL”, and “4 μg/mL”.
  11. Section “4.3. Statistical analysis” should be reconsidered. It is not necessary to show section 4.3.1 as an independent part.
  12. Conclusions need more in them. Please reconsider the conclusions carefully and give more perspectives on the conclusions.
Comments on the Quality of English Language

The quality of English needs improving. The manuscript has grammatical issues with improper syntax, spelling, punctuation, and other issues. However, the manuscript is not well written, and the language quality impairs the reader's understanding on several occasions. I recommend improving the manuscript with a language professional or a language polishing company.

Author Response

Dear Reviewer 2,

We would like to sincerely thank you for your review of our manuscript. Please find below our point-by-point response to your comments. We hope that the revisions we have made address your concerns satisfactorily in this revised version.

Best regards,

Maxim Kebenko

Round 2

Reviewer 1 Report

Comments and Suggestions for Authors

The manuscript "Downregulation of S6 kinase and Hedgehog-Gli1 by inhibition of fatty acid synthase in AML with FLT3-ITD mutation" presents small improvements but its statistical analysis is completely biased by using incorrect statistical tests that facilitate statistical significance (I mentioned this in my previous review). These corrections are absolutely necessary for it to be considered for publication.

The statistical analysis of your study is wrong, you are using Student's T-test (used when there are two groups) to study experiments with three or more groups (Figure 1 [3 groups], Figure 4 [3 groups], Figure 5 [3 groups], Figure 6 [3 groups], Figure 7 [3 groups], Figure 8 [3 groups], Figure 10 [10 groups]). An adequate statistical analysis with its corresponding post hoc is necessary to have reliability of the results.

Author Response

Dear Reviewer 1,

Please find attached our response to your comments.

Best regards,
Maxim Kebenko

Round 3

Reviewer 1 Report

Comments and Suggestions for Authors

The authors' response is insufficient and lacks an appropriate mathematical basis. Without proper analysis, their manuscript is not suitable for publication. The authors should perform an appropriate statistical analysis, in this case, an ANOVA with suitable post hoc for comparing the groups with a control.

Mathematical explanation:

Each t-test has a risk of making a Type I error (typically 5% when using α = 0.05). If you run multiple t-tests (e.g., A vs B, A vs C), the chance of making at least one Type I error increases rapidly.

  • Example: With 3 groups, there are 3 pairwise comparisons.

  • If each test has a 5% chance of error, the overall chance of making a false positive is more than 5%.

  • This problem is known as "inflated family-wise error rate.

Author Response

Dear Reviewer 1,

We would like to sincerely thank you for your review of our manuscript. Please find below our point-by-point response to your comments. We hope that the revisions we have made address your concerns satisfactorily in this revised version.

Best regards,

Maxim Kebenko

The authors' response is insufficient and lacks an appropriate mathematical basis. Without proper analysis, their manuscript is not suitable for publication. The authors should perform an appropriate statistical analysis, in this case, an ANOVA with suitable post hoc for comparing the groups with a control.

Mathematical explanation:

Each t-test has a risk of making a Type I error (typically 5% when using α = 0.05). If you run multiple t-tests (e.g., A vs B, A vs C), the chance of making at least one Type I error increases rapidly.

  • Example: With 3 groups, there are 3 pairwise comparisons.
  • If each test has a 5% chance of error, the overall chance of making a false positive is more than 5%.
  • This problem is known as "inflated family-wise error rate.

In addition to the required ANOVA analysis, and following extensive discussions with our team at the Institute of Statistics, we incorporated additional statistical tests to further support the validity of our findings. Statistical significance was assessed using one-way ANOVA followed by Tukey’s post hoc test for experiments involving more than two groups (minimum sample size: n = 3). For comparisons between two groups or when n < 3, unpaired Student’s t-tests were applied. Paired t-tests were used for specific within-group comparisons in experiments involving multiple datasets. For datasets with high variance, Cohen’s d was calculated to estimate effect size.

We included the statement regarding pS6 regulation in MOLM13 in the supplementary material due to the lack of statistical significance; this adjustment did not affect the overall conclusions of our experiments. The raw data, along with detailed statistical calculations, have also been provided. We hope these revisions address your proposal satisfactorily.

Round 4

Reviewer 1 Report

Comments and Suggestions for Authors

No additional comments